# CodeSwap: Symmetrically Face Swapping Based on Prior Codebook

Xiangyang Luo
goodluoxy@gmail.com
Guangdong Laboratory of Artificial
Intelligence and Digital Economy (SZ)
Shenzhen, Guangdong, China

Xin Zhang
zhangxin0526@stu.xjtu.edu.cn
Guangdong Laboratory of Artificial
Intelligence and Digital Economy (SZ)
Shenzhen, Guangdong, China

Yifan Xie
knownxyf@gmail.com
Guangdong Laboratory of Artificial
Intelligence and Digital Economy (SZ)
Shenzhen, Guangdong, China

Xinyi Tong
txy18@mails.tsinghua.edu.cn
Tsinghua Shenzhen International
Graduate School, Tsinghua University
Shenzhen, Guangdong, China

Weijiang Yu
weijiangyu8@gmail.com
School of Computer Science, Sun
Yat-sen University
Guangzhou, Guangdong, China

Heng Chang
changh17@tsinghua.org.cn
Tsinghua-Berkeley Shenzhen
Institute, Tsinghua University
Shenzhen, Guangdong, China

Fei Ma*
mafei@gml.ac.cn
Guangdong Laboratory of Artificial
Intelligence and Digital Economy (SZ)
Shenzhen, Guangdong, China

Fei Richard Yu
richard.yu@ieee.org
College of Computer Science and
Software Engineering, Shenzhen
University
Shenzhen, Guangdong, China
Carleton University
Ottawa, Canada

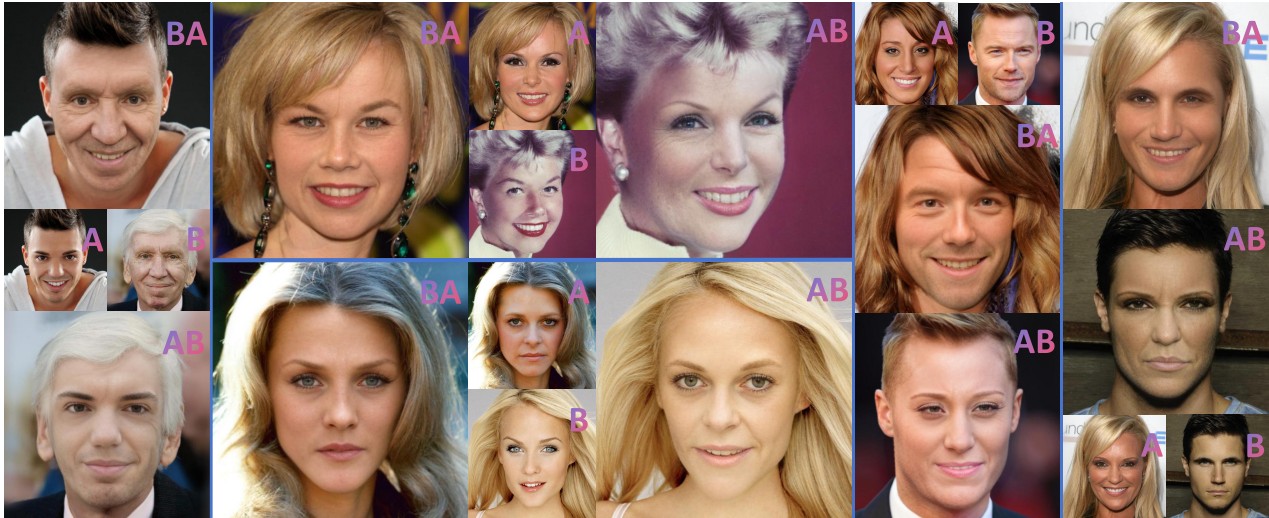

**Figure 1: Qualitative results of our model with $512 \times 512$ pixels. AB denotes the result of swapping A to B and BA denotes the result of swapping B to A. Benefiting from our powerful symmetrical framework, we can achieve face swapping results with high-fidelity and realism.**

*Corresponding Author.

## Abstract

Face swapping, the technique of transferring the identity from one face to another, merges as a field with significant practical applications. However, previous swapping methods often result in visible artifacts. To address this issue, in our paper, we propose *CodeSwap*, a

© 2024 Copyright held by the owner/author(s). Publication rights licensed to ACM.
ACM ISBN 979-8-4007-0686-8/24/10
https://doi.org/10.1145/3664647.3681120

symmetrical framework to achieve face swapping with high-fidelity and realism. Specifically, our method firstly utilizes a codebook that captures the knowledge of high quality facial features. Building on this foundation, the face swapping is then converted into the code manipulation task in a code space. To achieve this, we design a Transformer-based architecture to update each code independently, which enable more precise manipulations. Furthermore, we incorporate a mask generator to achieve seamless blending of the generated face with the background of target image. A distinctive characteristic of our method is its symmetrical approach to processing both target and source images, simultaneously extracting information from each to improve the quality of face swapping. This symmetry also simplifies the bidirectional exchange of faces in a singular operation. Through extensive experiments on ClelebA-HQ and FF++, our method is proven to not only achieve efficient identity transfer but also substantially reduce the visible artifacts.

## CCS Concepts

• **Computing methodologies → Image-based rendering**.

## Keywords

symmetrical face swap, code manipulation, facial prior, Transformer

**ACM Reference Format:**
Xiangyang Luo, Xin Zhang, Yifan Xie, Xinyi Tong, Weijiang Yu, Heng Chang, Fei Ma, and Fei Richard Yu. 2024. CodeSwap: Symmetrically Face Swapping Based on Prior Codebook. In *Proceedings of the 32nd ACM International Conference on Multimedia (MM '24), October 28-November 1, 2024, Melbourne, VIC, Australia.* ACM, New York, NY, USA, 10 pages. https://doi.org/10.1145/3664647.3681120

## 1 Introduction

With the rise of video media and the continuous development of image editing methods, face swapping techniques receive a lot of attention from both academic community and industry. With widely applications in various fields such as digital avatars [31], the movie industry, privacy preservation [6], face swapping aims to seamlessly transfer the identity of a source face to a target face while preserving the attributes of the target face, including expression, pose, and background.

Initially, face swapping methodologies were elementary, relying on simple cropping and replacement of inner facial pixels [3, 7]. These methods are often sensitive to variations in lighting and angles, resulting in distinguishing forgeries easily. The rise of Generative Adversarial Networks (GANs) significantly revolutionize this field. A variety of GAN-based approaches have been introduced [5, 11, 22, 46], proficient in transferring identity features of the source while retaining the attributes of the target face, but also struggle in generating high fidelity result. Additionally, there are also methods [23, 47, 52] utilizing pre-trained StyleGAN [19] to achieve a higher definition face swap results. And more recently, diffusion based methods for face swapping like those in [20, 50] have emerged. However, these methods still encounter challenges associated with visible artifacts.

To tackle this problem, we propose a novel framework called CodeSwap, which manipulating codes symmetrically in a code space to achieve face swap. Specifically, our method leverage a discrete codebook rich in high quality facial features, which can effectively eliminate visible artifacts and improve the generation quality. After encoding the images into the code space, the swapping task is transferred to meticulously identifying and substituting specific segments of the target face from the codebook to achieve face swapping. To achieve this, we design a global fusion network based on Transformers [39] to capture global information from both images and manipulate the codes within the code space. It is worth noting that our approach does not require an additional face recognition model to extract the identity features of the source image for explicit control. Instead, we design a Unified Self-Attention mechanism, which treat source and target images equally, enabling a capture of more information and mutual face swapping in a single step. Different from [24], which employs an asymmetric training method to stabilize the training process, our network is structurally symmetric and can not be decoupled.

Furthermore, we propose a mask generator module that utilize middle-layer semantic features of our encoder and decoder to generate masks and successfully ensures high quality image generation. Extensive experiments demonstrate that our method can generate swapping results with high-fidelity and realism. In summery, our contributions are as follows:

- We design a novel symmetrical face swapping structure by simultaneously and equally utilizing the information from two input facial images.
- We propose an innovative approach that conceptualizes face swapping as a process of code manipulation in a code space. By leveraging priors inherent to high quality faces, our method greatly reduces the occurrence of visible artifacts.
- Through extensive experiments, we demonstrate that our approach outperform the previous methods, especially in the term of high-fidelity and realism.

## 2 Related Work

### 2.1 Face Swapping

Research in face swapping attracts significant interest because of its potential practical applications. Early efforts in addressing this challenge primarily utilized classical image processing techniques and three-dimensional morphable models (3DMMs) [3, 29], which frequently resulted in face swaps that looked obviously artificial. However, the emergence of generative networks marked a significant step forward. FSGAN [29], an early pioneer, utilizes GANs for face reenactment and blending back to target image, but struggle with preserving the target's authentic attributes. This problem facilitates the development of AdaIN-based [15] methods [5, 11, 22], which focus on identity transfer and attribute preservation. Sim-Swap [5], for instance, adopts a pretrained face recognition model to extract the identity features of the source image and fuses them with the target features in the latent space, while FaceShifter [22] emphasizes attribute retention by incorporating target features during upsampling. InfoSwap [11] introduces mutual information to disentangle non-identity information, aiming for extracting the most valuable information from the identity representation. Blend-Face [35] proposes a new face encoder to mitigate attribute leakage. To further enhance image quality, several studies [25, 44, 45, 52] have explored StyleGAN's [19] potential for refined face swapping.

MegaFS [52] creates face swaps by substituting the target images' high-level semantic features with those from the source images. FSLSD [45] conveys attributes across multiple levels by utilizing the side-outputs from StyleGAN. E4S [25] adopts regional GAN inversion to separating facial shape and texture. Despite such advancements, the quest for artifact-free results continues. Recent diffusion-based methods [20, 50] introduce innovative approaches. For example, DiffFace [20] utilizes addtional facial guidance driven by pretrained models to guide the denoising process, and DiffSwap employs a 3DMM to blend faces and extract landmarks for explicit facial shape control. Although these diffusion-based methods adopt more complex training procedures and extended inference durations, the problem of visible artifacts remain. However, our method suppresses visible artifacts to a large extent, resulting in a more natural-looking face swap.

## 2.2 Codebook Learning

Traditional methods that adopt sparse representation through the use of learned dictionaries have significantly advanced image processing tasks, particularly in the areas of restoration [12, 36, 37, 48] and denoising [9]. Building on these works, innovations like Vector Quantized Variational AutoEncoders (VQVAE) [32, 38] and Vector Quantized Generative Adversarial Networks (VQGAN) [10] take a further step by introducing the idea of compact codebooks. These codebooks efficiently compress image data into a set of discrete vectors, resulting in a compact and rich image representation. This development has significantly raised the bar for image quality and computational efficiency, offering improvements over traditional sparse representation methods by facilitating more detailed image reconstructions with less computational overhead. Such progress finds extensive application across various domains, including face restoration [13, 42, 51], motion generation [27, 43], gesture synthesis [2], and so on.

Drawing inspiration from the concept of codebook learning, this work utilizes a prior codebook of faces to achieve face swapping. By manipulating codes with high quality facial priors, we can achieve more natural face-swapping results.

## 3 Method

The overarching structure of our face swapping framework is illustrated in Figure 2. This section intoduces our innovative approach to face swapping. We begin by describing the training of prior codebook in Section 3.1, which serves as the foundation of our framework. Leveraging this facial prior, we create a globally aware and symmetric code swapping framework. This framework effectively update the original facial code with entries from the codebook in the code space, facilitating precise face swapping. For further details, please refer to Section 3.2. To preserve the complex background details, we introduce a mask generator in Section 3.3. This module utilizes middle layer features of encoder and decoder to generate masks that seamlessly integrate with our face swapping architecture.

## 3.1 Stage 1: Facial Prior Codebook Learning

The face prior guarantees the generation of highly natural faces with diversity. To capture the facial prior knowledge, we leverage

the task of face reconstruction through large number of natural faces as depicted in Figure 3, utilizing the VQGAN [10] architecture. The facial image $I_f \in \mathbb{R}^{H \times W \times 3}$ undergoes a transformation into a code space via an encoder $\mathcal{E}$, yielding $Z_f \in \mathbb{R}^{m \times n \times d}$. Subsequently, within this code space, we substitute each item by identifying the nearest term from a learnable codebook $Q = \{q_k \in \mathbb{R}^d\}_{k=1}^N$, thereby acquiring the quantized facial features $Z_q \in \mathbb{R}^{m \times n \times d}$ :

$$Z_q = (\underset{q_k \in Q}{\text{argmin}}\{||Z_{ij} - q_k||_2\}) \in \mathbb{R}^{m \times n \times d} \tag{1}$$

Finally, $Z_q$ is sent to decoder $\mathcal{D}$ to obtain the restructed face image $\hat{I}_f$. As shown in the lower path in Figure 3, given a low quality face image, the prior codebook learned from natural faces can reduce the visible artifacts and produce a high quality recovered one.

*3.1.1 Training Losses.* Similar to [10, 38], we adopt gradient replication to address the issue of gradient truncation that arises during the quantization process. This strategy allows us to train the encoder, decoder and prior codebook in an end-to-end manner utilizing the following loss function:

$$L = L_1 + L_{per} + \lambda L_{adv} + L_{code}, \tag{2}$$

where the first three of terms aims to enhance the quality of the reconstruction. Here, $L_1$ represents the L1 loss, $L_{per}$ indicates perceptual loss [17], and $L_{adv}$ denotes adversarial loss with $\lambda$ serving as a adaptive weight. These components are expressed as follows:

$$L_1 = ||I_f - \hat{I}_f||_1, L_{per} = ||VGG(I_f) - VGG(\hat{I}_f)||_2,$$
$$L_{adv} = \log D(I_f) + \log(1 - D(\hat{I}_f)). \tag{3}$$

The last term, $L_{code}$, is meticulously crafted to minimize the discrepancy between the code $Q$ and the features $Z_f$, defined as:

$$L_{code} = ||sg[Z_q] - Z_f||_2 + \beta||sg[Z_f] - Z_q||_2, \tag{4}$$

where $sg[\cdot]$ stands for gradient-stop operation and $\beta = 0.25$ is a weighting factor.

## 3.2 Stage 2: Symmetrically Code Swapping

After training the a prior codebook, the next step is to manipulate the code of the image to achieve face swapping. As depicted in Figure 2, face images $I_a$ and $I_b$ are fed into a shared encoder to the code space and then flattened to derive the feature embeddings $Z^a, Z^b \in \mathbb{R}^{mn \times d}$. These embeddings are then concatenated to form $Z$:

$$Z = Concat(Z^a, Z^b) \in \mathbb{R}^{2mn \times d}. \tag{5}$$

Since the encoder is trained on the reconstruction task in Stage 1, the concatenated feature $Z$ contains all the information of the two images, which we called global features. The global features $Z$ will be fed into a global fusion module and a code selector to choose the best codes from prior codebook to achieve face swap.

**Global Fusion Module** is consist of several unified self-attention mechanism, which can fuse the features of two images with a global attention and output new embeddings symmetrically. The fusion mechanism is shown in Figure 4 (b) and expressed as follows:

$$Attention(Z_l) = Softmax(\frac{Q_l K_l^T}{\sqrt{d_k}})V_l = AV_l, \tag{6}$$

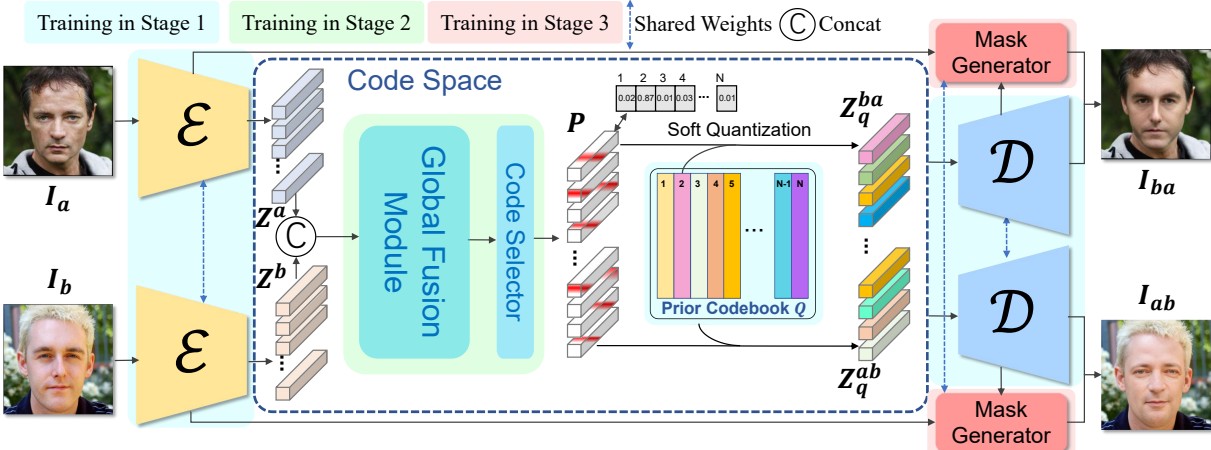

**Figure 2: The symmetrical architecture of our CodeSwap. The encoder $\mathcal{E}$ , decoder $\mathcal{D}$, and prior codebook $Q$ in the blue background are trained first (Section 3.1), then we train our Transformer-based global fusion module in the green background (Section 3.2), and the mask generator in the red background is trained in Stage 3 (Section 3.3).**

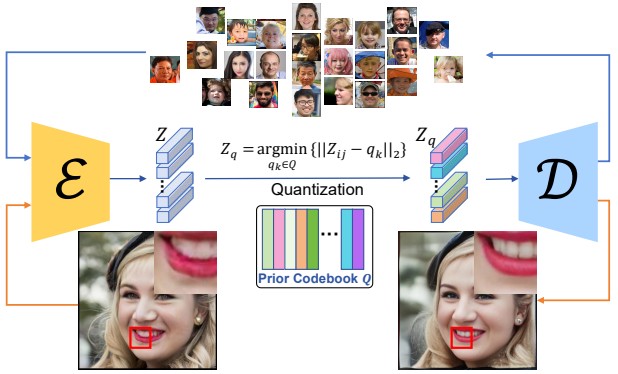

**Figure 3: The upper path demonstrate the trainning phase of prior codebook based on natural face images. And the lower path shows our high quality facial prior can reduce the visible artifacts of low quality images.**

where $Z_l$ denotes the output of the $l^{th}$ layer, $Q_l, K_l, V_l$ are obtained from $Z_l$ through linear transformations from $W_Q \in \mathbb{R}^{d \times d_k}$, $W_K \in \mathbb{R}^{d \times d_k}$, $W_V \in \mathbb{R}^{d \times d_v}$, respectively. And $A$ is the calculated attention map of all codes from both images. After the self-attention mechanism, a linear layer is utilized to project the dimension into $2mn \times d$, and a Feed-Forward Network (FFN) to generate the updated features $Z_{l+1}$.

As shown in Figure 4, compare to traditional AdaIN-based methods, our global fusion module has several advantages.

i) Previous AdaIN-based methods [5, 11, 22] use additional pre-trained face recognition models to extract identity features from the source image. While some methods employ more powerful encoders to gather extensive information [49, 52], they may overlook the potential contributions of the target image. In contrast, our model uses an encoder from a reconstruction task to extract global

features from both images and applies unified self-attention to leverage information from both, thus enhancing the swap process.

ii) Compared to AdaIN's impact on the overall style of the image, our approach manipulates each code individually based on the calculated attention map, which ensures more precise updating of features.

iii) Our symmetric process manipulates the features of both images at the same time, thus enabling swapping two faces at once, extending a functionality absent in previous methods.

**Code Selector.** After the global fusion module, we adopt a code selector to select the suitable items from the prior codebook to update the original one, which is a MLP module and can be expressed as:

$$P = \text{MLP}(Z_{last}) \in \mathbb{R}^{2mn \times N}, \tag{7}$$

where $Z_{last} \in \mathbb{R}^{2mn \times d}$ is the output of global fusion module and $N$ denotes the number of items in codebook. Instead of employing *argmax* function for $P$ to get the code with the highest probability, we adopt soft quantization to refer to all the codes in the codebook, which is expressed as:

$$Z_q = Softmax(P) \times Q \in \mathbb{R}^{2mn \times d}. \tag{8}$$

The obtained $Z_q$ selectively refers to the individual codes and contribute to the generation of more natural details, which is then passed to the decoder $\mathcal{D}$ to generates high-fidelity swapped face images.

*3.2.1 Training Losses.* To train our symmetrical swap module effectively, we apply following losses:

$$L_{swap} = L_{recon} + \lambda_{id}L_{id} + \lambda_{per}L_{per} + \lambda_{exp}L_{exp} + L_{cyc}, \tag{9}$$

where the weights $\lambda_{id}, \lambda_{per}, \lambda_{exp}$ are set to 1.5, $2 \times 10^{-3}$, and 0.5, respectively. The perceptual loss, $L_{per}$, is same to Equation (3). The identity loss, $L_{id}$, ensures the transfer of identity features between face images and is expressed as:

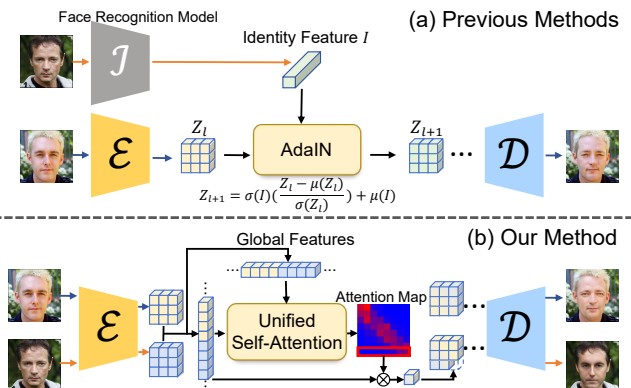

**Figure 4: Comparison between our global fusion method and previous AdaIN-based feature fusion methods. (a) Previous Methods [5, 11, 22, 35] fuse the whole features through AdaIN. (b) Our self-attention based approach has two advances: 1) we are able to accept more feature information from both images, 2) we update each code individually based on calculated attention map for more precise control.**

$$L_{id} = 1 - \frac{CosSim(E_{id}(I_a), E_{id}(I_{ab})) + CosSim(E_{id}(I_b), E_{id}(I_{ba}))}{2}, \quad (10)$$

where $E_{id}$ represents a pre-trained face recognition model [8] and $CosSim(\cdot, \cdot)$ denotes the cosine similarity between two vectors. The reconstruction loss, $L_{recon}$, is applied to ensure fidelity when the input faces $I_a$ and $I_b$ in the training sample are identical. It is formulated as:

$$L_{recon} = \begin{cases} \frac{||I_{ba} - I_a||_1 + ||I_{ab} - I_b||_1}{2} & \text{if } I_a = I_b \\ 0 & \text{otherwise} \end{cases} \quad (11)$$

What's more, $L_{exp}$ aims control ensure the resulting expression is consistent with the face being swapped [26], expressed as:

$$L_{exp} = \frac{||E_{exp}(I_a) - E_{exp}(I_{ba})||_2 + ||E_{exp}(I_b) - E_{exp}(I_{ab})||_2}{2}. \quad (12)$$

Our symmetrical architecture also effectively utilizes code cycle loss $L_{cyc}$, which enhances the stability of training. Due to the frozen encoder and decoder in this stage, we do not need to rerun the full network, instead we only focus on achieving this loss in the code space. $L_{cyc}$ can be formulated as:

$$L_{cyc} = ||Z_q^{aba} - Z^a||_2 + ||Z_q^{bab} - Z^b||_2, \quad (13)$$

where embeddings $Z_q^{ba}$ and $Z_q^{ab}$ are the result codes in the first swap, and they are processed again through the modules in Stage 2 to derive $Z_q^{aba}$, $Z_q^{bab}$. Comparing $Z_q^{aba}$, $Z_q^{bab}$ against original embeddings $Z^a$, $Z^b$ for reconstruction loss, we can stable the training process to produce high-fidelity results. This comprehensive loss $L_{swap}$ ensures the successful transfer of identity features, while maintaining the fidelity and expression consistency of the swapped faces.

## 3.3 Stage 3: Mask Generator

In order to keep all the background information of the target image, we design a mask generator that can distinguish between identity-specific and unrelated regions. Inspired by previous methods [1, 44], our approach utilize the middle layer features from both encoder and decoder of our framework to produce detailed masks. This involves concatenating features from encoder and decoder, refining through $1 \times 1$ convolutions, activations, and upsampling to generate the final mask, which is detailed in the appendix. The blending operation yields the final images as follows:

$$I_{ab} = I_{ab}^{naive} \odot M_b + I_b \odot (1 - M_b), \quad (14)$$

where $\odot$ denotes Hadamard product.

*3.3.1 Training Losses.* For optimal mask generator training, we employ a dual-strategy approach: initially adopting a supervised method by pretrained parsing model for coarse mask generation, followed by an unsupervised technique to finetune the mask generertor for fine-grained mask creation. During the supervised phase, We utilize ground truth obtained by a pretrained parsing model from input images to guide mask production, with the training loss formulated as:

$$L = ||M - M_{GT}||_1. \quad (15)$$

Subsequently, we integrate the mask generator into the face swapping framework in Stage 2, leveraging losses in Stage 2 for unsupervised training. This enables the mask generator to automatically distinguish between areas that are relevant and irrelevant to the identity feature.

## 4 Experiment

### 4.1 Datasets

We train our models using two high-definition facial datasets: FFHQ [19], which comprises 70,000 facial images, and VGGFace2-HQ [30], which is a high resolution version of VGGFace2 [4] and contains millions of images with thousands of identities. Both datasets are widely recognized and utilized for training generative models due to their diverse and high quality image content. To train our model, we resize the images into $512 \times 512$. To better validate the effect of high resolution face swapping, we process the CelebA-HQ [18] and FaceForensics++ (FF++) [33] as our test set, which represent high quality images and low quality images, respectively.

### 4.2 Experimental settings

Our model processes $512 \times 512$ images within a $16 \times 16 \times 512$ latent space and utilizes a codebook of 1,024 codes, following the autoencoder structure from [10]. Our model features a 6-layer Transformer and is trained across three stages: 1,000K iterations for Stage 1, 600K for Stage 2, and 10K for Stage 3, using a global batch size of 32 on two NVIDIA Tesla A100 GPUs. We employ the Adam optimizer [21] with a $10^{-4}$ learning rate.

### 4.3 Comparison with previous methods

To demonstrate the effectiveness of our method, we conduct quantitative experiments, qualitative experiments and user study, comparing our method with FSGAN [28], Simswap [5], InfoSwap [11],

**Table 1: Comparison of face swapping methods on CelebA-HQ and FF++ for ID retrieval, ID similarity, pose error, expression error, and Frechet Inception Distance. Bold text highlights the best scores, and underline for the second best scores. The results demonstrate that our method achieves comparable results to existing mothods.**

| Method | CelebA-HQ | | | | | FF++ | | | | |
|---|---|---|---|---|---|---|---|---|---|---|
| | ID R. ↑ | ID Sim. ↑ | Expr. ↓ | Pose. ↓ | FID ↓ | ID R. ↑ | ID Sim. ↑ | Expr. ↓ | Pose. ↓ | FID ↓ |
| FSGAN [28] | 24.2 | 0.1934 | 0.0341 | 0.0336 | 54.05 | 31.56 | 0.2510 | 0.0315 | 0.0212 | 15.36 |
| SimSwap [5] | 97.8 | 0.5990 | 0.0346 | 0.0339 | 30.18 | 92.99 | 0.5488 | 0.0355 | 0.0149 | 7.48 |
| MegaFS [52] | 77.1 | 0.4188 | 0.0414 | 0.1577 | 53.10 | 50.70 | 0.2899 | 0.0404 | 0.1202 | 28.96 |
| InfoSwap [11] | 94.5 | 0.5860 | 0.0354 | 0.0316 | 21.67 | 86.81 | 0.4945 | 0.0396 | 0.0280 | 12.46 |
| DiffSwap [50] | 17.2 | 0.2570 | **0.0227** | 0.0223 | 23.18 | 18.80 | 0.2158 | **0.0247** | 0.0170 | 11.86 |
| BlendFace [35] | 89.6 | 0.5158 | 0.0260 | **0.168** | 20.92 | 78.48 | 0.4358 | 0.0311 | 0.0135 | **3.84** |
| E4S [25] | 87.7 | 0.4873 | 0.0385 | 0.0887 | 24.78 | 86.82 | 0.4870 | 0.0422 | 0.0550 | 25.34 |
| CodeSwap | **98.7** | **0.6118** | 0.0292 | 0.0187 | **20.68** | **93.72** | **0.5571** | 0.0301 | **0.0130** | 8.90 |

**Figure 5: Qualitative comparison with other methods. Our method not only achieves excellent identity transfer but also generates images with high-fidelity and realism. (Zoom in for details.)**

MegaFS [52], DiffSwap [50], BlendFace [35], and E4S [25] as baselines.

*4.3.1 Quantitative Comparisons.* In quantitative study, we utilize two distinct test datasets: CelebA-HQ [18] and FF++ [33]. For the CelebA-HQ dataset, we select a subset of 1,000 images featuring

unique identities and craft 1,000 testing pairs through random selection. In the case of FF++, we extract 10 random frames from each video, resulting in a dataset of 10,000 images. These images are then combined randomly to generate an additional 10,000 images for testing.

We benchmark our method against prior works, evaluating performance across several metrics including ID retrieval, ID similarity, pose and expression accuracy, and the Frechet Inception Distance (FID) [14]. ID retrieval is performed using a specialized face recognition model [41] to extract identity features, employing cosine similarity for matching. For CelebA-HQ, we match images within the selected subset, while for FF++, to heighten the challenge, we adopt the approach of [23] by matching against a single frame from each video instead of all frames. Pose accuracy is assessed through the Euclidean distance between estimated and ground truth poses [34]. Meanwhile, expression accuracy is evaluated by comparing the L2 distance between expression embeddings from the swapped and target faces using a model distinct from the one used during the training phase [40].

As shown in Table 1, our method obtains the highest ID Retrieval and ID Similarity metrics on both datasets, indicating that our method works well to transfer the facial identity from source to target. The expression and pose accuracy metrics show that we are also able to preserve the attribute of the target face. In addition to this, our method achieves the lowest FID on the CelebA-HQ dataset , indicating the capability to generate high-fidelity images of the proposed methods. Our FID scores are slightly higher on the FF++ dataset, which is likely due to visible artifact reduction in our generated images compared to the original FF++ images. This leads to differences in data distribution and the increase of FID.

*4.3.2 Qualitative Comparisons.* We perform qualitative comparisons on CelebA-HQ [18] and FFHQ [19]. As shown in Figure 5, existing methods often have visible artifacts, especially at the corners of the mouth, nose, and forehead. But our method greatly suppresses visible artifacts, producing a more natural image with a higher clarity. It is noteworthy to mention the observation made in the last line of Figure 5. When there is a significant difference in the mouth shapes between the source and the target, existing approaches either failing to alter the mouth shape properly or resulting in obvious visible artifacts. In contrast, our method correctly changes the shape of the mouth, renders the skin in the constricted area of the mouth, and blends the skin color of the target face with the texture of the source face for a natural look.

*4.3.3 User Study.* To explore human perception of face swapping results, we perform a user study encompassing three distinct evaluations with ratings on a scale from 1 to 5: 1) identity similarity to the source image, 2) consistency of expression, lighting, and pose with the target image, and 3) the degree of realism. As illustrated in Table 2, our method outperforms existing methods in all metrics, especially in realism. These findings suggest that our method effectively preserves the attributes of the target image and ensures a high-fidelity generation of realistic images, all while successfully transfer the identity from the perspective of human.

### Table 2: User Study Comparison.

| Method | Consistency ↑ | Attribute ↑ | Realism ↑ |
|---|---|---|---|
| FSGAN [28] | 1.82 | 2.68 | 2.05 |
| SimSwap [5] | 2.88 | 2.73 | 2.27 |
| MegaFS [52] | 1.86 | 1.36 | 1.86 |
| InfoSwap [11] | 2.86 | 3.05 | 2.35 |
| BlendFace [35] | 2.96 | 3.36 | 3.04 |
| DiffSwap [50] | 2.19 | 2.55 | 2.18 |
| E4S [25] | 3.58 | 2.14 | 2.68 |
| CodeSwap | **3.63** | **3.56** | **3.41** |

## 4.4 Ablation study

In this section, we prove the validity of some of the modules in our CodeSwap.

*4.4.1 Effectiveness of Prior Knowledge via the Codebook.* To assess the critical function of the codebook in embedding prior knowledge, we skip the code selector and prior codebook, allowing the Transformer's feature outputs to directly proceed to the decoder and retrain the model. The qualitative and quantitative results are shown as w/o $C$ in Figure 6 and Table 3, respectively. The qualitative results, illustrated in Figure 6, demonstrate that the lack of codebook results in more artifacts in the generated image and a blurrier face. The FID metrics in Table 3 also demonstrates the bad quality of generated images. These results prove the important role of the prior codebook in generating high-fidelity images.

*4.4.2 Significance of Symmetrical Swap.* In our approach, we employ symmetrical swap, which utilize self-attention on concatenated features to obtain the global information of both images and achieve face swapping with each other at once. To prove the significance of symmetrical swap, we adopt the cross-attention mechanism to fuse the features from the source face, thus breaking the symmetrical process. The qualitative and quantitative results are shown as w/o $S$ in Figure 6 and Table 3, respectively. Qualitative results show that the generated images are not precise enough in expression accuracy especially the shape of the mouth. Quantitative experiments also shows the degradation of its generation quality. Both results demonstrate the validity of our symmetrical framework.

*4.4.3 Impact of Soft Quantization.* In our approach, we utilize the output probabilities from the code selector to achieve soft quantization through a weighted averaging process. To evaluate the efficacy of this technique, we contrasted two quantization methods: 1) hard quantization which select the code with the highest probability instead of our weighted average, and 2) Gumbel-Softmax trick [16], which correspond to w/o $S_1$ and w/o $S_2$ both in Figure 6 and Table 3, respectively. The quantitative results of Table 3 show that both methods have varying degrees of degradation in their effectiveness especially the gumbel-softmax trick. The qualitative results demonstrate that the use of soft quantization is able to synthesize the information from the individual codes, leading to more natural details.

*4.4.4 Impact of the Mask Generator.* Our CodeSwap focuses on facial priors to generate high quality faces. In addition, we design a mask generator to preserve the background information of the

**Table 3: Quantitative result of ablation study on CelebA-HQ.**

| Method | ID R ↑ | ID Sim ↑ | Exp ↓ | Pose ↓ | FID ↓ |
|--------|--------|----------|-------|--------|-------|
| w/o $C$ | 98.1 | 0.5788 | 0.0319 | 0.0229 | 26.22 |
| w/o $S$ | 98.3 | 0.5859 | 0.0327 | 0.0225 | 20.77 |
| w/o $Q_1$ | 97.7 | 0.5606 | 0.0299 | 0.0218 | 20.93 |
| w/o $Q_2$ | 86.7 | 0.4271 | 0.0313 | 0.0369 | 20.96 |
| Ours | **98.7** | **0.6118** | **0.0292** | **0.0187** | **20.68** |

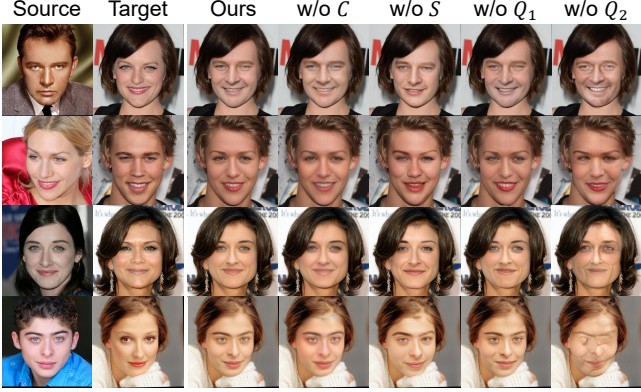

**Figure 6: Qualitative ablation results of CodeSwap.**

target image. To validate the impact of our mask generator, we conduct the experiment under the setting without mask generator (w/o $M$). The result is shown in Figure 7, which demonstrates that our generated mask can distinguish the identity related area and help to achieve better background preservation.

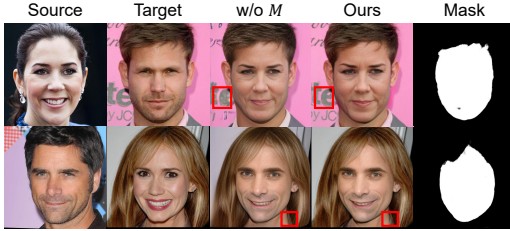

**Figure 7: Qualitative ablation study on mask generator.**

## 4.5 Additional Study

In this section, we extend our application to regional control face swap and analyze some of the additional results.

*4.5.1 Regional Control Swap.* Although our method manipulates on all codes of the input image, we demonstrate the ability of regional control swap of our method. This can be achieved through selective manipulation of codes in the code space. As shown in Figure 8, our method allows for the high quality regional face swap, which is not possible with purely pixel-level operations. This characteristic ensures that even if part of the codes are manipulated, the resulting image blends perfectly with other features of the face, achieving highly natural results.

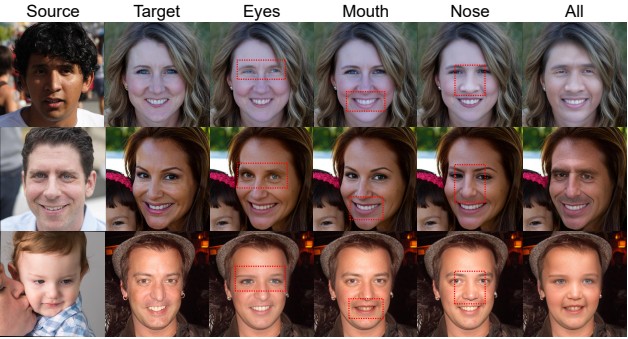

**Figure 8: Regional swap by only manipulates the code of the corresponding area.**

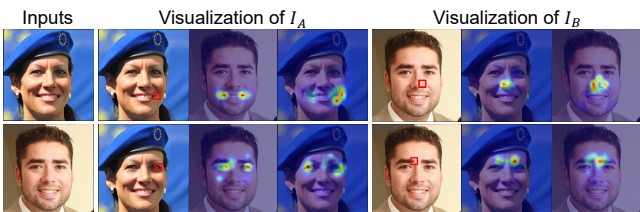

**Figure 9: Visualization of attention maps for a given area and each region has attention on both input images for symmetrical swapping.**

*4.5.2 Visualization of Attention Map.* To gain a deeper insight into the feature fusion process of our global fusion module, we visualize the attention maps generated during self-attention, as depicted in Figure 9. Since we perform the self-attention mechanism after concatenating the features of the two images, each code obtains two attention maps from the image itself and the other input image. The visualization results show that the global fusion module not only successfully focuses on correct region of the other image but also on other regions of its own image. This proves that our symmetrical framework successfully obtains information on both images and automatically discovers an optimal feature fusion strategy that is particularly beneficial for face swap tasks.

## 5 Conclusion

In this paper, we propose the concept of code manipulation as a novel approach to achieve face swapping. By leveraging a prior codebook learned from natural face images, we manipulate the codes to achieve high-fidelity results. Unlike traditional techniques for extracting identity features, we propose a symmetrical framework that concentrates on the global information of the both images and manipulates each code individually to achieve precise face swap of two images at once. Our method is validated across a broad range of experiments, proving its superior ability to produce natural, high-fidelity facial images. We believe that this approach will significantly contribute to advancing the field of face swapping research.

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
