# OpenReview forum: "CodeSwap: Symmetrically Face Swapping Based on Prior Codebook"
_acmmm.org/ACMMM/2024/Conference — MM2024 Poster_

### Official Review · Reviewer_i1AZ · 2024-05-14

**Rating:** 4
**Confidence:** 4

**Summary:**

This paper introduces a symmetric face-swapping framework that utilizes vector quantization techniques. The framework concentrates on the global information of the two images and manipulates each code separately, allowing for precise face swapping. The effectiveness of this approach is validated through qualitative and quantitative experiments.

**Strengths:**

1. The paper is well-written and easy to understand.
2. The paper visualizes results well and has other applications.

**Limitations:**

1. Why did unidirectional face swapping previously lead to the creation of artifacts? What experiments support this conclusion?
2. How are fine-grained masks produced? If a pre-trained model was used to generate the Ground Truth, why not use this model directly instead of adding an extra step to auto-generate the mask?
3. Why need to use vector quantization techniques?
4. The current problem with face-swapping is its poor performance when the gap between face shapes is large. The authors' experiments do not appear to highlight the advantages in this respect.

**Suitability:**

3

---

### Official Review · Reviewer_sEve · 2024-05-24

**Rating:** 3
**Confidence:** 3

**Summary:**

This paper leverages a codebook to capture high-quality facial features. The method converts face swapping into a code manipulation task using a Transformer-based architecture. Additionally, a mask generator is incorporated to seamlessly blend the swapped face with the background of the target image.

**Strengths:**

1. This paper proposes an innovative approach that conceptualizes face swapping as a process of code manipulation in a code space.
2. The method has the ability of regional control swap through selective manipulation of codes in the code space.

**Limitations:**

1. The paper does not compare this method with face-swapping methods based on StyleGAN. StyleGAN-based face-swapping methods can generate highly realistic facial images.
2. The paper mentions that a mask generator is used to achieve seamless blending of the generated face with the background of the target image, but it does not provide specific challenging examples of face occlusion to illustrate the advantages of the algorithm. Provide examples where faces are partially occluded (e.g., by hair, hands, glasses, etc.), showing how your algorithm successfully achieves seamless blending.

**Suitability:**

3

---

### Official Review · Reviewer_LNoe · 2024-05-24

**Rating:** 4
**Confidence:** 4

**Summary:**

This paper presents a Transformer-based architecture designed to independently update each code, enabling more precise manipulations. Extensive experiments on CelebA-HQ and FF++ demonstrate that the method not only achieves efficient identity transfer but also significantly diminishes visible artifacts.

**Strengths:**

1. This paper introduces an intriguing approach that conceptualizes face swapping as a manipulation of codes within a designated code space.
2. In addition to metric tests, this paper also conducted a user study.
3. The authors performed extensive ablation experiments.

**Limitations:**

1. Why are the representations learned in the GAN's latent space limited, while those in the code space are not? It is suggested that the authors provide further explanation.
2. How are the test pairs on FF++ constructed, and why are they chosen randomly instead of using the original dataset's test pairs?

**Suitability:**

3

---

### Official Review · Reviewer_gjBy · 2024-05-25

**Rating:** 3
**Confidence:** 4

**Summary:**

This paper presents a symmetrical framework to achieve face swapping with high-fidelity and realism. A Transformer-based architecture is proposed to update face codes, which enables precise face swapping manipulations. Meanwhile, a mask generator is incorporated to achieve seamless blending of the generated face with the background of target image. Extensive experiments on face datasets verify the ability of the proposed method to not only achieve efficient identity transfer but also substantially reduce the visible artifacts.

**Strengths:**

1. The architecture details of each proposed module are described in detail and the figures illustrates each module well, which are easy to follow.
2. The authors provide rich experiments on different datasets, e.g., CelebA-HQ and FF++. Experimental results show the priority of the proposed method over other face swapping baselines from both quantitative and qualitative aspects.

**Limitations:**

1. The novelty or the first contribution of this paper that states about the symmetrical design is quite limited. The previous paper [1] has already present this design and this paper was posted to arXiv in 2022. And the description in this paper does not adequately explain the motivation for symmetrical design and its advantages.
2. The analysis of the limitations of the diffusion methods is insufficient. The authors should provide a more in-depth explanation or experimental verification of why limited representations or unidirectional face swapping design can lead to synthesis artifacts.
3. What’s the main differences between Stage 1 and the training process of VQGAN? Why not adopt the VQGAN pre-trained on the FFHQ dataset?
4. For the Regional Control Swap in section 4.5.1, how the selective manipulation of codes in the code space can be achieved to enable such fine-grained facial feature control?
5. Lack of comparative analysis of the related face swapping work that is also based on attention mechanism for code swapping [2].

[1] Q. Li, W. Wang, C. Xu, Z. Sun and M. -H. Yang, "Learning Disentangled Representation for One-Shot Progressive Face Swapping," in IEEE Transactions on Pattern Analysis and Machine Intelligence, doi: 10.1109/TPAMI.2024.3404334.
[2] Zeng H, Zhang W, Fan C, et al. Flowface: Semantic flow-guided shape-aware face swapping[C]//Proceedings of the AAAI Conference on Artificial Intelligence. 2023, 37(3): 3367-3375.

**Suitability:**

3

---

### Meta-Review · Area_Chair_jw7s · 2024-06-30

**Recommendation:** Accept (Poster)
**Confidence:** 3

**Metareview:**

The paper received mixed scores (BR,WR,BA,BA). One reviewer decreased the final rating (BA->WR) yet without deepened reasoning, while another increased their score (BR->BA) after rebuttal clarified some mistakes in the initial review.

The scores do not provide a very clear and unique indication. The major positive aspects of the paper are the extensive experiments which show the method outperforms previous ones, and the conceptual idea of casting the problem of face swapping as manipulation of pieces of latent codes in a disentangled latent space. Whereas this idea is not innovative as pointed out by gjBy, it seems to be addressed in a way such that disentanglement is achieved locally, so that swapping results are improved quantitatively and qualitatively.

Most limitations found by reviewers were clarified in the rebuttal. Some still remain, in particular the lack of comparison with some recent methods and explanations as pointed out by gjBy, and lack of technical novelty.

Even though some comparisons are neglected and some deepened analyses missing, the overall contribution is still considered good.

Considering the reviewers' scores, comments, paper and rebuttal, the AC preliminarily still recommends acceptance. On the one hand reviewers found some technical similarities with previous works ([1] in gjBy review), yet the rebuttal provided quite detailed description of the conceptual differences and advantages. On the other hand, the positive comments from reviewers suggest such changes led to an improvement in the performance. Overall, whereas the paper does not introduce breakthrough discoveries it looks a good contribution.